

# Modelling earthquake rates and associated uncertainties in the Marmara Region, Turkey

Thomas Chartier[1,2,*], Oona Scotti[2], Hélène Lyon-Caen[1], Keith Richard-Dinger[3], James H. Dieterich[3], and Bruce E. Shaw[4]

[1]Ecole Normale Supérieure, PSL University, CNRS, UMR 8538 - Laboratoire de Géologie, 24 rue Lhomond, 75005 Paris, France
[2]Bureau d'Evaluation des Risques Sismiques pour la Sûreté des Installations, IRSN, Fontenay-aux-Roses, 92262, France
[3]Department of Earth Sciences, University of California, Riverside, 92521, California, USA
[4]Lamont Doherty Earth Observatory, Columbia University, Palisades, NY 10025,USA
[*]now at GEM Hazard Team, GEM Foundation, via Ferrata, 1, 27100 Pavia, Italy

**Correspondence:** Thomas Chartier (thomas.chartier@globalquakemodel.org)

**Abstract.** Modelling the seismic potential of active faults and the associated epistemic uncertainty is a fundamental step of probabilistic seismic hazard assessment (PSHA). We use SHERIFS (Seismic Hazard and Earthquake Rate In Fault Systems), an open-source code allowing to build hazard models including earthquake ruptures involving several faults, to model the seismicity rates on the North Anatolian Fault (NAF) system in the Marmara region. Through an iterative approach, SHERIFS converts the slip-rate on the faults into earthquake rates that follow a Magnitude Frequency Distribution (MFD) defined at the fault system level, allowing to model complex multi-fault ruptures and off-fault seismicity while exploring the underlying epistemic uncertainties. In a logic tree, we explore uncertainties concerning the locking state of the NAF in the Marmara Sea, the maximum possible rupture in the system, the shape of the MFD and the ratio of off-fault seismicity. The branches of the logic tree are weighted according to the match between the modelled earthquake rate and the earthquake rates calculated from the local data, earthquake catalogue and paleoseismicity. In addition, we use the result of the physics-based earthquake simulator RSQSim to inform the logic tree and increase the weight on the hypotheses that are compatible with the result of the simulator. Using both the local data and the simulator to weight the logic tree branches, we are able to reduce the uncertainties affecting the earthquake rates in the Marmara region. The weighted logic tree of models built in this study is used in a companion article to calculate the probability of collapse of a building in Istanbul.

## 1 Introduction

The North Anatolian Fault System (NAFS) runs through the North of Turkey for a distance of more than 1500 km (Figure 1) accommodating the westward extrusion of the Anatolian plate with a right lateral motion of around 25 mm/yr (e.g. Reilinger et al. (2006)). Along the western portion of the NAFS, this right lateral motion is partitioned between two fault branches with

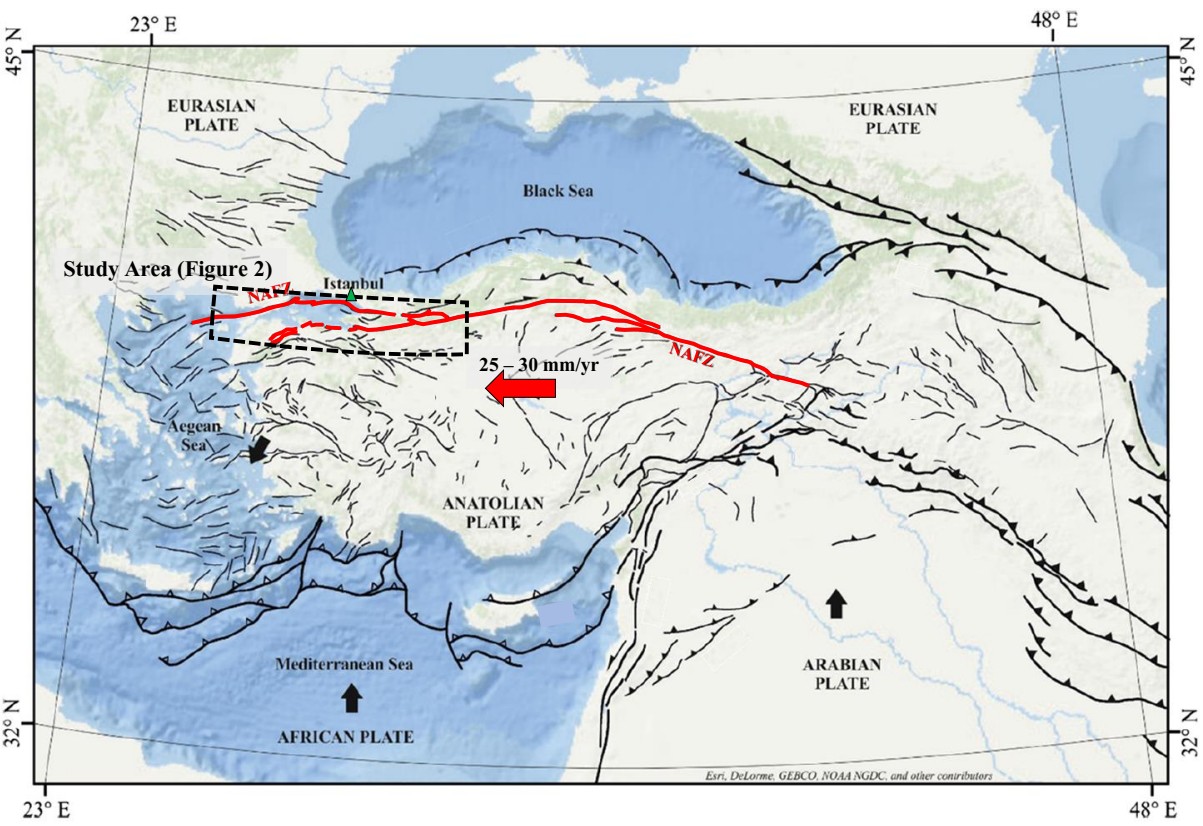

**Figure 1.** Regional tectonic setting. Modified from Duman et al. (2018). The active faults are in black except for the North Anatolian Fault Zone (NAFZ) which is in red. The black and red arrows show the plate motion relative to a fixed Eurasia. The dotted box indicates the study area detailed in figure 2.

the northern branch accommodating most of the motion (Reilinger et al. (2006)). This northern branch of the NAFS crosses the Sea of Marmara 20 km south of the city of Istanbul (Figure 2). The last earthquakes having occurred in the Marmara Sea, severely damaging the city of Istanbul, date back to 1766 and 1894 (Ambraseys (2002)). Growing from a population of less than one million in 1950, Istanbul is now a city of more than fifteen million. The Mw=7.6 1999 Izmit earthquake that ruptured the fault eastward of the Marmara Sea has been a reminder of the hazard faced by Istanbul. The NAFS is expected to be the

major source of seismic hazard looming on the megalopolis. Motivated by better constraining the seismic hazard, the two decades following the 1999 earthquake have been rich in geological, geophysical and geodetic studies of the Marmara Region. These studies have brought new light on the structure (Armijo et al. (1999); Le Pichon et al. (2001); Le Pichon et al. (2003)) and active deformation (Hergert and Heidbach (2010); Ergintav et al. (2014); Bohnhoff et al. (2017)) of the NAFS and raised important questions such as the locking condition of the fault and the possibility of a large rupture in the Marmara Sea.



Building on the findings of these studies, several Earthquake Rate Forecast (ERF) models and hazard maps have been developed (Erdik et al. (2004); Gülerce et al. (2017); Murru et al. (2016); Sesetyan et al. (2018); Demircioğlu et al. (2018)). While these studies improve our understanding of the hazard for Istanbul, none of them fully explore the epistemic uncertainties in the parametrization of the ERF parameters that have the potential to affect the seismic hazard in Istanbul. The NAFS being offshore on its portion closer to Istanbul, satellite-based geodesy (GPS and INSAR) are not able to resolve accurately the

locking condition of the fault which remains debated (Klein et al. (2017)). The possibility of creep in the Marmara Sea might have an impact on the seismic hazard for Istanbul and should be explored in the seismic hazard assessment.

Erdik et al. (2004) and Demircioğlu et al. (2018) have used the classic approach to model faults and background seismicity in seismic hazard. The rate of earthquakes in the background is based on the local seismic catalogue up to a threshold magnitude (M5.5) and the rate of larger magnitude earthquakes is based on the slip-rate of the faults. This approach can lead to several

limitations in Probabilistic Seismic Hazard Assessment (PSHA) and eventually in the seismic risk assessment. First, this approach lacks the consideration that earthquakes with a magnitude larger than the threshold magnitude can occur in the background as it has been observed in July 2019 in California where a magnitude 7.1 earthquake occurred on a fault that was not modelled as active in the UCERF3 model (Brandenberg et al. (2019); Field et al. (2014)) but the possibility of such an earthquake occurring in the background was considered. Second, this approach when used with a spatially uniform distribution

of the seismicity, cannot consider that intermediate magnitude earthquakes (lower than the threshold magnitude) can be more likely to occur on the fault than on any given point of the background.

In this study, we use the recently developed SHERIFS code (Chartier et al. (2019)) that allows exploring these uncertainties using a logic tree approach. Furthermore, SHERIFS considers a system level approach, in which different rupture scenarios are explored in an aleatory manner, relaxing fault segmentation and defining complex multi-faults ruptures. Through the SHERIFS

approach we can also address two issues (1) the uncertainty on the size of the largest rupture that may occur along the NAFS and (2) the shape of the Magnitude Frequency Distribution (MFD).

The geometry of the network could potentially host larger ruptures than the one observed in the historical time. For example, the change of azimuth in the geometry of the fault in front of the Princes Island (Figure 2) that was not crossed by the 1999 Izmit earthquake could be crossed by a future earthquake as it has been shown by dynamic rupture modelling (Oglesby et al.

(2008); Aochi and Ulrich (2015)). Furthermore, ruptures that run through more complex fault geometries have been observed in fault systems (e.g. the 2016 Kaikoura earthquake, Klinger et al. (2018)).

It has been long observed that the number of earthquakes in a region decreases with the magnitude of the earthquake. Gutenberg and Richter (1954) established the logarithmic decrease of the number of earthquakes with the magnitude (noted as GR law hereafter). However, the possibility for the seismicity on individual faults to follow a different type of MFD has

been discussed, notably in California where several studies have discussed this aspect of earthquake statistics along the San Andreas strike slip fault systems; arguing either for the GR law (Page and Felzer (2015)) or for a discrepancy (Schwartz and Coppersmith (1984)). Recently, Stirling and Gerstenberger (2018) have analysed several fault zones in New Zealand and have argued for the systematic exploration of the uncertainty on the shape of the MFD when modelling faults in seismic hazard assessment. We explore this uncertainty in this study.



The earthquake rates modelled with SHERIFS using each combination of uncertainties will be compared to the earthquake rates calculated from the earthquake catalogue and the paleoseismic records in order to give a score to each model. However, for some hypotheses, the comparison with the data is not sufficient for rating one hypothesis against another. We tackle this issue by modelling a synthetic catalogue of the fault system using the earthquake simulator RSQSim (Richards-Dinger and Dieterich (2012)). By analysing the statistics of the synthetic catalogue, we are able to discuss the physical validity of the

different hypotheses explored and give more weight to the branches of the logic tree that are more compatible with physically-based models. Because the aim of this study is to better understand the seismic risk in Istanbul, presented in a companion paper (ref), the uncertainties concerning the modelling of the earthquake rates in the fault system are propagated to calculate the probability of collapse of a building in 50 years. The impact of each uncertainty on the uncertainty of the estimation of the probability of collapse is quantified and discussed.

## 2    Model parameters for the Western North Anatolian Fault System

Fault traces and some of the slip rate estimates along the North Anatolian fault system have been recently updated in Emre et al. (2018) and references within. In this study, we rely on their map (available online at http://www.mta.gov.tr/v3.0/hizmetler/ yenilenmis-diri-fay-haritalari, last accessed August 2019) to digitise the fault traces (Figure 2).

    The parameters of the main faults used in this study are presented in Table 1. Slip rates are taken from Hergert and Heidbach

(2010) who inverted the GPS velocity field in a geomechanical model in order to calculate the slip-rate of the fault network. These slip-rates are in general agreement with the slip-rates calculated by Flerit et al. (2004) and Ergintav et al. (2014). Since they agree with the long-term geological slip-rates of the faults, these values of slip-rates have been preferred to those estimated by Reilinger et al. (2006) calculated at a larger scale because they tend to overestimate the slip-rates compared to the geological estimates (Hergert and Heidbach (2010)). Uncertainties on slip-rates (Table 1) are taken into account by a random exploration

in a uniform distribution between the minimal slip-rate value and the maximal slip-rate value set from the literature (Hergert and Heidbach (2010); Klein et al. (2017)) and weighted in order to give a most weight to the mean value. It can be noted that the range of slip-rates explored in this study excludes values that vastly differ from the geological estimate (such as the 23 mm/yr according to e.g. Le Pichon et al. (2003)), since they integrate the deformation on a wider area than the fault itself.

    The scientific community has been debating over the possibility of the NAF been creeping in the Western Marmara region,

along the Terkidag, Central basin, Kumburgaz and Acvilar sections of the fault (table 1, figure 2). Creep has been observed on other strike-slip faults in the world as for San Andreas Fault (Nason (1973)) or the Longitudinal Fault Taiwan ( Hsu and Bürgmann (2006)) and is expected to release an important part of the tectonic load of the fault without producing major earthquakes, hence possibly reducing the seismic hazard in the region. The large scientific efforts of the past two decades have led to conclude that the Eastern part of the fault in the Marmara Sea is locked (Diao et al. (2016)). However, the distance from

the fault to the shore in the western part of the sea remains a challenge for traditional land-based or satellite-based geodetic instruments. While novel sea-floor geodesy experiments seem to suggest that the fault is locked at the surface at least at some places (Lange et al. (2019)), these conclusions are still under discussion (Yamamoto et al. (2019)). Relocated microseismicity

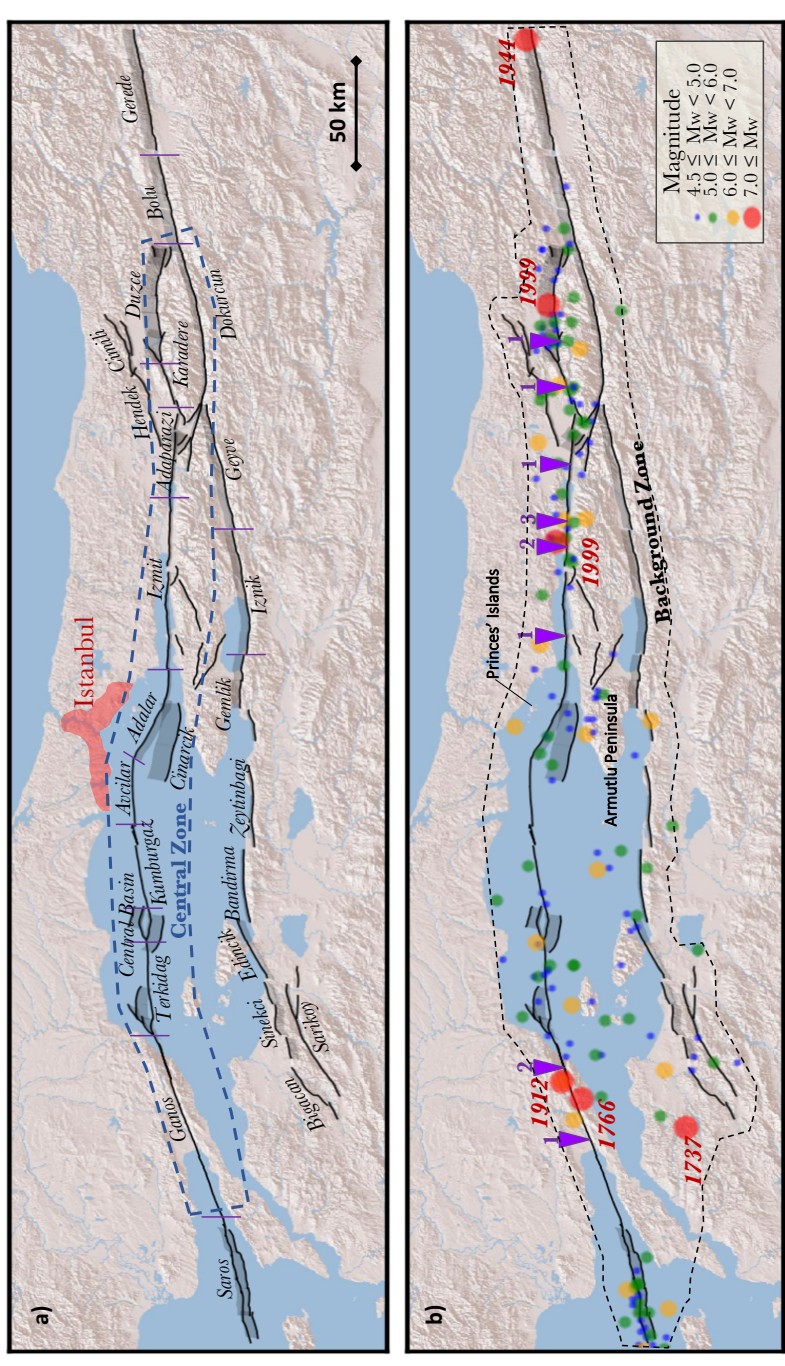

**Figure 2.** Fault system and the earthquake catalogue used in this study (Emre et al. (2018)). a) The faults sections are delimited by the purple lines and the name of the fault sections are indicated in black. b) Earthquakes of magnitude greater than magnitude 4.5 and during the complete period (see Table 2) in the Marmara Region. The year of occurrence of the largest earthquakes (M>7) are indicated. The yellow triangles indicate the locations of paleoseismic studies and the numbers reference the Table 3.



| Fault Name | Rake | Creep hypothesis | Seismogenic depth (km) | Min slip-rate (mm/yr) | Mean Slip-rate (mm/yr) | Max slip-rate (mm/yr) |
|---|---|---|---|---|---|---|
| Ganos | 180 | Fully Locked | 15 | 17 | 18 | 19 |
| Terkidag | 180 | Creeping | 15 | 0. | 0. | 0. |
| | | Partial Creep | 15 | 7.5 | 8. | 8.5 |
| | | Deep Creep | 5 | 17 | 18 | 19 |
| | | Fully Locked | 15 | 17 | 18 | 19 |
| Central Basin | 180 | Creeping | 15 | 0. | 0. | 0. |
| | | Partial Creep | 15 | 7.5 | 8. | 8.5 |
| | | Deep Creep | 5 | 17 | 18 | 19 |
| | | Fully Locked | 15 | 17 | 18 | 19 |
| Kumburgaz | 180 | Creeping | 15 | 0. | 0. | 0. |
| | | Partial Creep | 15 | 7.5 | 8. | 8.5 |
| | | Deep Creep | 15 | 17 | 18 | 19 |
| | | Fully Locked | 15 | 17 | 18 | 19 |
| Acvilar | 180 | Creeping | 15 | 7.5 | 8. | 8.5 |
| | | Partial Creep | 15 | 7.5 | 8. | 8.5 |
| | | Deep Creep | 15 | 17 | 18 | 19 |
| | | Fully Locked | 15 | 17 | 18 | 19 |
| Adalar | -168 | Fully Locked | 10 | 12.2 | 14.3 | 16.5 |
| Cinarcik | -147 | Fully Locked | 15 | 2.2 | 3.6 | 5 |
| Izmit | 180 | Fully Locked | 15 | 18 | 20 | 22 |
| Adaparazi | 180 | Fully Locked | 15 | 16 | 18 | 20 |
| Karadere | 180 | Fully Locked | 15 | 11.8 | 15 | 18 |
| Duzce | 180 | Fully Locked | 15 | 11.8 | 15 | 18.2 |

**Table 1.** Model parameters of the faults of the Marmara region, closer to Istanbul. The full parameters table for all the faults in the model is available in the electronic supplement. See the text for details on slip rates setting and definition of partial and deep creep.

and the identification of repeater earthquakes suggest that the lower part of the fault might be creeping (Schmittbuhl et al. (2016b); Schmittbuhl et al. (2016a)). Based on the latest studies of the fault in this area, we propose to explore four models of

locking condition for the NAF that represent the current state of knowledge. The "creep" hypothesis considers the fault as fully creeping, thus reducing its contribution to the seismic slip-rate budget to 0. The "partial creep" hypothesis considers the fault releasing half of its slip-rate as creep and half as seismic moment, reducing its slip-rate by half. The "deep creep" hypothesis considers that the fault is fully locked for the first 5 kilometers and creeping below, and the "fully locked" hypothesis considers that 100% of the fault slip-rate can be released as earthquakes.


For the faults within the Armutlu peninsula (figure 2), we were not able to find slip-rate estimates. However, the velocity field shows very little deformation (Ergintav et al. (2014)) within the Armutlu peninsula. Based on this observation, we assume the slip-rate of these faults to be less than 1 mm/yr. It is worth noting that these faults are relatively far from Istanbul in comparison to the NAF and their participation in the seismic hazard affecting the city is expected to be negligible.

        In this study, we combined two catalogues : the earthquake catalogue from Kadirioğlu et al. (2018) homogenised in Mw
for the period 1900-2012 and the catalogue SHEEC (Stucchi et al. (2013)) for the period 1000-1899. The completeness period used in this study are based on these two studies and presented in Table 2.

| Completeness magnitudes | 4.0 - 4.7 | 4.8 - 5.2 | 5.3 - 5.7 | 5.8 - 6.2 | 6.3 - 6.7 | 6.8 - 7.2 | 7.3 - 7.7 | 7.8+ |
|---|---|---|---|---|---|---|---|---|
| SHARE (Woessner et al. (2015)) | 1987 | 1952 | 1900 | 1850 | 1750 | 1700 | 1700 | 1700 |
| Kadirioğlu et al. (2018) | 1992 | 1977 | 1937 | 1850 | 1750 | 1700 | 1700 | 1700 |

**Table 2.** Completeness time as a function of magnitudes used in this study.

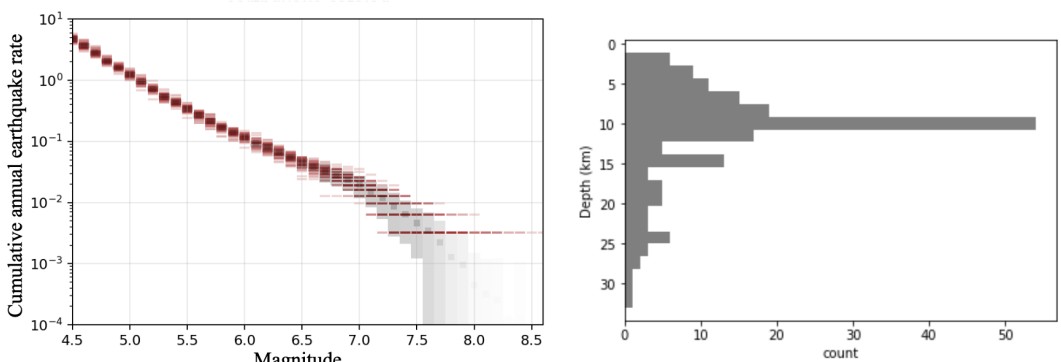

**Figure 3.** Left : Magnitude Frequency Distribution for the study area calculated from the catalogue and the completeness periods presented in Table 2. The red dashes are individual Monte Carlo samples on the earthquake magnitude uncertainties and the completeness period. The mean rate value is indicated by black squares and the 16th and 84th percentiles are indicated by the grey area. Right : Depth distribution of earthquakes of magnitude 4.5 and above in the background zone Kadirioğlu et al. (2018) since 1992.

        Based on the analysis of the depth distribution of earthquakes in the instrumental catalogue (Kadirioğlu et al. (2018)), 80% of the earthquakes are in the first 15km of the crust in the region of interest (Figure 3). Furthermore, slip inversion of the Izmit earthquake show that most of the slip during the earthquake was contained in the first 15km of the crust (Reilinger et al.
(2000)). Thus in this study we have limited the seismogenic thickness of the fault system to a depth of 15 km with the exception of the Adalar section of the fault stopping at 10km deep in order to avoid crossing the Cinarcik fault at depth.

        Dikbaş et al. (2018) present a review of the paleoseismic studies that have been carried out on the Izmit, Adapazari, Karadere and Duzce segments of the fault. They propose an interpretation of the rupture history of these segments. Based on the number



| Site | Reference | Number of events | Observation time (years) | Annual earthquake rate | Uncertainty |
|------|-----------|------------------|--------------------------|------------------------|-------------|
| Ganos 1 | Rockwell et al. (2009) | 4 | 1000 | 0.00357 | 0.001 |
| Ganos 2 | Meghraoui et al. (2012) | 5 | 1600 | 0.003 | 0.001 |
| Izmit 1 | Klinger et al. (2003) | 4 | 1600 | 0.0025 | 0.0013 |
| Izmit 2 | Dikbaş et al. (2018) | 5 | 1600 | 0.0032 | 0.0014 |
| Izmit 3 | Dikbaş et al. (2018) | 5 | 1600 | 0.0032 | 0.0014 |
| Adaparazi 1 | Dikbaş et al. (2018) | 3 | 1600 | 0.0018 | 0.001 |
| Karadere 1 | Dikbaş and Akyüz (2011) | 3 | 1600 | 0.0018 | 0.001 |
| Duzce 1 | Pantosti et al. (2008) | 4 | 1600 | 0.0025 | 0.0013 |

**Table 3.** Annual rate of M 7.2+ earthquakes on sections of the NAF deduced from paleoseismic studies.

of ruptures observed in one location and the length of the observation period, we estimate the annual rate of earthquakes
with a magnitude larger or equal to 7.2 +- 0.1 for different sections of these segments. The location of these segments are
presented in figure 2 and the estimated rates are presented in table 3. At the West of the Marmara Sea, for the Ganos segment,
the paleoearthquake rates have been calculated from Rockwell et al. (2009) and Meghraoui et al. (2012). Both studies lead to
similar rates for earthquakes larger than 7.2 (table 3). The paleoearthquake rates are calculated by dividing the number of events
by the observation time, the uncertainty reflects the statistical uncertainty due to the small number of observation assuming a
poissonian process.

## 3 Earthquake rate modelling

In this section, we describe two approaches for calculating earthquake rates in fault systems. The first approach is SHERIFS
(Chartier et al. (2019)), a statistical approach that converts the slip-rate of the faults into earthquake rates built to follow a
given shape of MFD at the system level. The second approach is RSQSim that generates a long catalogue of synthetic ruptures
using the loading rates and the rate and state equation (Richards-Dinger and Dieterich (2012)). SHERIFS allows to explore a
wide range of epistemic uncertainties on the input hypotheses and RSQSim allows to discuss which of these hypotheses are
physically plausible.

### 3.1 The statistical approach: SHERIFS

#### 3.1.1 Core principle and main input hypothesis

SHERIFS uses an iterative budget spending approach of the slip-rate of the fault to calculate the annual rate of occurrence of
each rupture of a predefined set of ruptures. In an iterative manner, SHERIFS randomly selects user-defined rupture scenarios
for which involved faults have slip-rates budget to spend. The random selection is done in order to ensure that the resulting
system level MFD has the shape imposed as input (b value in the case of a Gutenberg-Richter MFD, for example). It is im-



| Magnitude | 4.0 | 4.5 | 5.0 | 5.5 | 6.0 | 6.5 | 7.0 | 7.5 | 8.0+ |
|---|---|---|---|---|---|---|---|---|---|
| Background 1 | 0.8 | 0.8 | 0.8 | 0.8 | 0.8 | 0.9 | 1.0 | 1.0 | 1.0 |
| Background 2 | 0.7 | 0.7 | 0.7 | 0.7 | 0.7 | 0.8 | 0.9 | 1.0 | 1.0 |
| Background 3 | 0.4 | 0.4 | 0.4 | 0.45 | 0.5 | 0.6 | 0.8 | 1.0 | 1.0 |

**Table 4.** Ratio of earthquakes assumed to be on the faults over the total number of earthquakes in the system, for each background hypothesis.

portant to recall that SHERIFS doesn't simulate earthquakes but only converts slip-rates into earthquake rates. For this reason, SHERIFS is computationally light which allows aleatory (combination of ruptures) and epistemic (logic tree) uncertainties concerning the fault system to be easily explored.

SHERIFS takes as input the geometry and slip-rate of the faults, the set of multi-fault ruptures that can be expected in the fault network, and the shape of the MFD defined at the fault system level. Before the calculation, the actual value of the MFD and the shape of the MFD of each individual fault are not known. They will be deduced from the fault slip-rate budget and the other hypotheses. Depending on the combination of input hypotheses and fault parameters, SHERIFS can consider part of the slip-rate budget of some faults as Non Main Shock (NMS) slip in order to respect the target MFD shape. A NMS of more than 30% is most likely an indication that the combination of input hypotheses used doesn't agree with the fault parameters in the SHERIFS framework and that they should be reconsidered.

### 3.1.2 Background seismicity

One uncertainty that SHERIFS allows to explore is the proportion of seismicity that can occur in the background, on faults that are unknown or not considered as active in the model. In most PSHA, this is taken into account by a background zone with a GR MFD truncated at a given Mt.

In SHERIFS, it is possible to define a-priori the proportion of earthquakes that can be expected on the faults and the proportion in the background for each range of magnitude. In order to assess these proportions, we analyze the distance of each earthquake to the closest fault in the model. Considering the poor knowledge on the epicentral location of the historical earthquakes, we only consider the instrumental catalogue after 1970. Since we are interested in the hazard in Istanbul, we only consider the spatial distribution of earthquakes around the sea of Marmara (the central zone in Figure 2).

In order to represent the epistemic uncertainty associated with the proportion of seismicity considered on the faults versus that on the background, we set up three branches of the logic tree corresponding to three different background hypothesis (Figure 4,Table 4). The proportion of earthquakes considered to occur on the faults for each branch is presented in . Studies of the NAFS off-fault deformation might bring some light on these value in the future (Şengör and Zabcı (2019)).

### 3.1.3 MFD

The annual rates of earthquakes obtained by analysis of the Kadirioğlu et al. (2018) catalogue in the Central Zone (Figure 2) indicates a MFD diverging from a GR MFD for magnitudes larger than 6.5 (figure 3). The rates are obtained while exploring





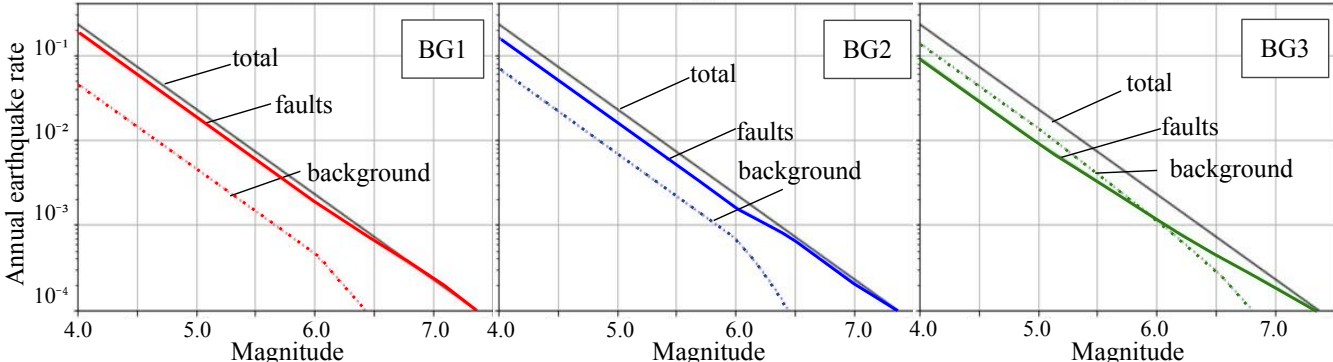

**Figure 4.** Distribution of earthquakes between the background seismicity and the faults for each background hypothesis. In black, the earthquake rate for the entire fault system (faults + background), the solid colour line is the faults only and the dashed line is the background only.

the uncertainties on the magnitude of earthquakes, on the completeness period length using a Monte Carlo approach and the uncertainties linked to the low number of earthquakes for the larger magnitudes. Despite this large number of uncertainties explored, the resulting MFD significantly differs from the GR MFD shape. We can observe a larger number of earthquakes of magnitude 7.0 and above than predicted by the GR law. Like any earthquake catalogue, the Turkish catalogue is short in comparison with the seismic cycle. Only 6 earthquakes larger than magnitude 7.0 are in the catalogue, and it can be argued

that the observed MFD is a result of the incomplete sample of the phenomenon and that the long term MFD should follow a GR distribution. This debate on the catalogue MFD concerns the Californian earthquake catalogue as well (Page and Felzer (2015),Parsons et al. (2018)).

In order to reflect the lack of consensus in the community, we explore two hypotheses for the target MFD : one where the target MFD follows a GR truncated between a minimum magnitude and a maximum magnitude, one where the target MFD follows a shape tuned to that of the rates deduced from the catalogue. The tuned shape MFD is described by the following equation and is composed of two parts, both defined by a double truncated GR with a b value deduced from the lower magnitude part of the catalogue (4.5 < M < 6.0) that follows an exponential decrease. This formulation of the MFD has been developed especially to closely resemble the MFD observed in the earthquake catalogue of our study area (Figure 3). If other formulations could also have been explored, this set of equations was found appropriate for the expected use in this study : matching the rates observed in the earthquake catalogue while exploring the uncertainty on the set of possible rupture scenarios. While we do not argument for the universal nature of this formulation, variations around this formulations could be useful for other study





regions.

$$P_i(M) = \begin{cases} \frac{e^{-\beta \times (M-0.05-M_{min})}}{1-e^{-\beta \times (6.8-M_{min})}} - \frac{e^{-\beta \times (M+0.05-M_{min})}}{1-e^{-\beta \times (6.8-M_{min})}}, & \text{if } M < 6.6 \\ P_i(m=6.6)/3, & \text{if } M = 6.7 \text{ or } 6.6 \\ P_i(m=6.3), & \text{if } M = 6.8 \\ P_i(m=6.8) \times (10^{(-b \times (M-6.8))}), & \text{if } M > 6.8 \end{cases}$$

where $P_i$ is the density function depending on $M$; $b$ describes the linear decrease with $M$; $\beta = b \times \ln 10$; and $M_{min}$ is the minimal magnitude considered in the calculation.

### 3.1.4 Set of rupture scenario

The historical catalogue doesn't contain any earthquake with magnitude larger than 7.5 since 1700. Based on the statistics of the earthquake catalogue, Bohnhoff et al. (2016) consider this value of 7.5 to be the largest possible magnitude in the Marmara region while earthquakes can reach up to magnitude 8.0 in the eastern part of the NAFS. A recent review of paleoseismological studies and historical earthquake studies (Dikbaş et al. (2018)) suggests that the largest earthquake having occurred in the region could have been a rupture including both the 1999 Izmit earthquake rupture area and the 1999 Duzce earthquake rupture area resulting in a magnitude 7.7 earthquake. We define a Set 1 of more than 300 Fault to Fault (FtF) ruptures allowing a large diversity of ruptures to be possible in the system with the largest magnitude of 7.7 (Figure 5a).

The resolution of the earthquake records diminishing as we consider older events, it is possible that larger earthquakes occurred in the NAFS but may not have been observed neither in the historical times nor in the paleoseismic records. Furthermore, since most bends in the system could be crossed by a rupture without jumping a large distance or large changes in azimuth, we need to imagine that larger earthquakes might be possible. This hypothesis has been considered in previous studies (Murru et al. (2016), Mignan et al. (2015)). We thus build a Set 2 of FtF ruptures to explore this possibility in which the largest possible ruptures correspond to a magnitude 8.0 earthquake (Figure 5b).

### 3.1.5 Locking condition of the NAFS in the Western Marmara region

As exposed during the presentation of the NAFS earlier, there is uncertainty concerning the locking condition of the NAF is the western Marmara Sea. Four hypotheses of locking conditions ranging from fully creeping to fully locked are explored in a logic tree to represent the current state of knowledge 1.

### 3.1.6 The logic tree

SHERIFS is run with the hypotheses of each of the branch of the logic tree (Figure 6). We then compare the modelled annual earthquake rates with the seismicity rates from the earthquake catalogue and the paleoseismic records.


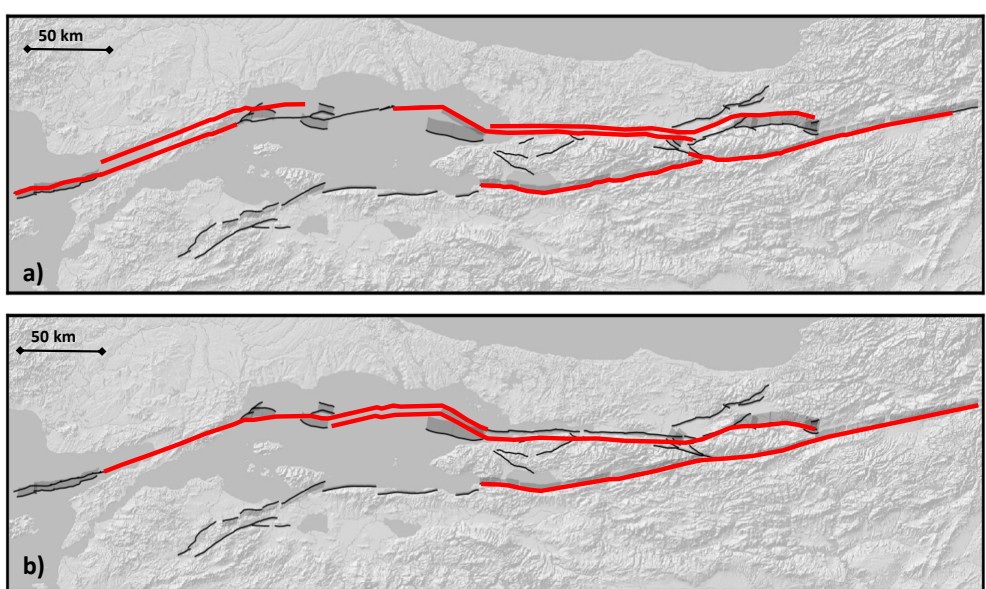

**Figure 5.** Example of the largest ruptures included in the models using the Set 1 (a) and the Set 2 (b). The full list of ruptures is available in the electronic supplement.

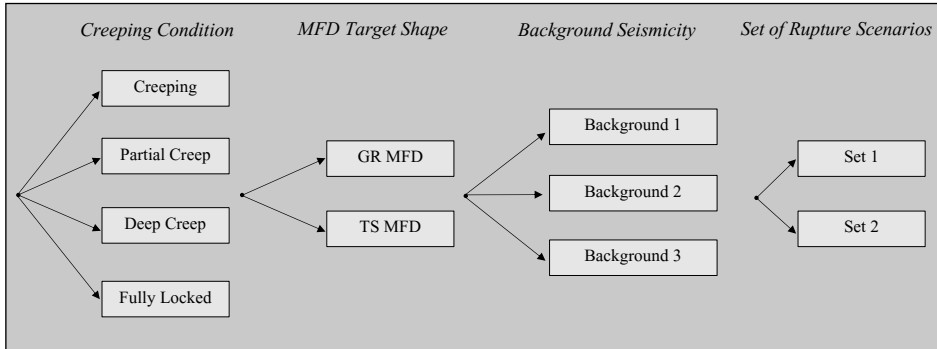

**Figure 6.** Logic tree explored in this study. For each branch, the scaling law parameters and the slip-rate uncertainties are explored through 10 random samples.




### 3.1.7 Results : Earthquake rates modelled by SHERIFS

In this study, we only modelled part of the NAF (inside the dotted box in figure 1), therefore some boundary effects might occur on the faults that would rupture with faults outside of the modelled zone. With the aim of modelling the seismic risk in Istanbul, far from the system boundaries, we compare only the rates for a region smaller than the whole modelled system centred on

Istanbul (Figure 2) that is not affected by boundary effects. In this zone, we extract the earthquakes from the catalogue and the modelled rate of ruptures. For ruptures that are only partly located in the zone, the rate is corrected by taking into account the proportion of the fault in the zone.

The rates modelled using SHERIFS are slightly higher than the ones of the catalogue in the central zone (figure 7). However, some combinations of hypotheses better fit the rate from the catalogue than others. The branches using the GR target shape

lead to a rate of earthquakes of magnitude between 4.5 and 6.5 that is larger than the observed rate for all models (figure 7). The tuned shape (TS) hypothesis is in better agreement with the catalogue for this range of magnitudes. Both MFD hypotheses reproduce the rate of large earthquakes equally well. The branches using the Set 2 of rupture scenarios, allowing ruptures up to magnitude 8, have lower earthquake rates than the branches using Set 1 and they obtain a better fit with the rates from the catalogue. The 4 different creep hypotheses lead to similar earthquake rates at the level of the central zone (figure 7). In

conclusion, the branches using the combination of the TS MFD and the Set 2 of ruptures est fit the rates calculated from the earthquake catalogue.

In Figure 8, we compare the participation rate of fault sections where an estimate of the rate of earthquakes is available based on paleoseismic studies. The participation rate is the rate of all the ruptures for which the given fault section is involved in. For most sites, the rate of earthquakes modelled is in general agreement with the rate deduced from the paleoseismicity for

all branches of the logic tree.

At the Ganos 1 site (Figure 9), the branch "Creep", assuming a complete creep in the west of the Marmara Sea, leads to rupture rates significantly lower than the paleoseismic rates. For this branch of the logic tree, the Ganos fault spends around 46% of its slip-rate budget as Non Main Shock (NMS) slip. The proportion of NMS slip is the proportion of slip-rate budget that couldn't be spent in seismic moment rates in SHERIFS. A large NMS (>20-30%) indicates the inability in SHERIFS to

satisfy the hypotheses of MFD, slip-rate budget and the set of ruptures. The introduction of a fault completely creeping in the western part of the Marmara Sea limits the number of large earthquakes along the Ganos fault. In the SHERIFS framework, the ability of the Ganos fault to release seismic moment jointly with the faults of the Marmara Sea is required in order for it to spend its full budget.

## 3.2 The physics-based approach : RSQSim

### 3.2.1 Core principle and main hypotheses

The comparison between the modelled earthquake rates and the data allows to reduce part of the uncertainties explored in the logic tree but some uncertainties still remain, notably the uncertainties concerning the MFD shape and the maximum rupture size.

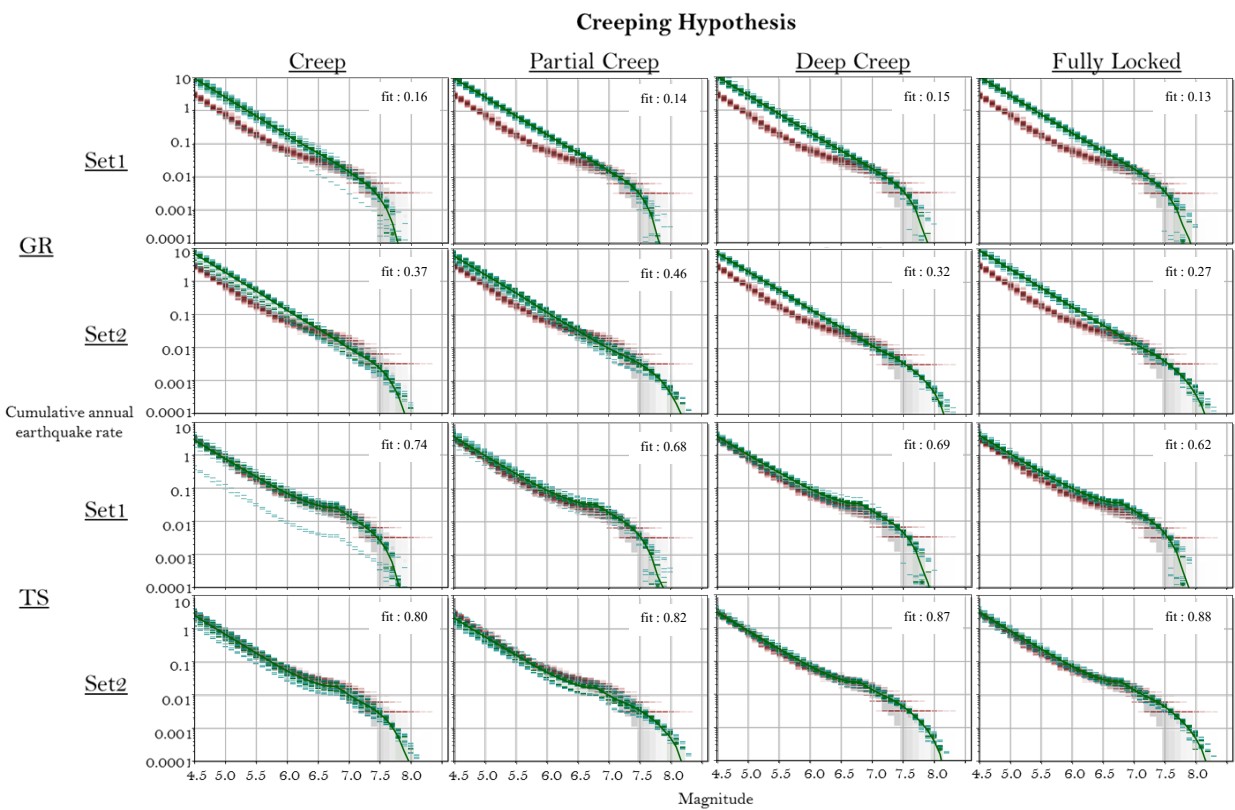

**Figure 7.** Comparison between the rates calculated from the earthquake catalog (in red with uncertainties in grey) and the model rates (in green) for the central zone of the fault system, close to Istanbul (figure 2). The dashed green lines are the MFD for each individual model of the logic tree. The dotted green line are the 16th and 84th percentiles of the distribution and the continuous green line is the mean value of the distribution. Each figure shows the match between model and data for each branch of the logic tree, organised as a table. A fit of 1. is a perfect fit and a value close to 0. expresses a very poor fit.


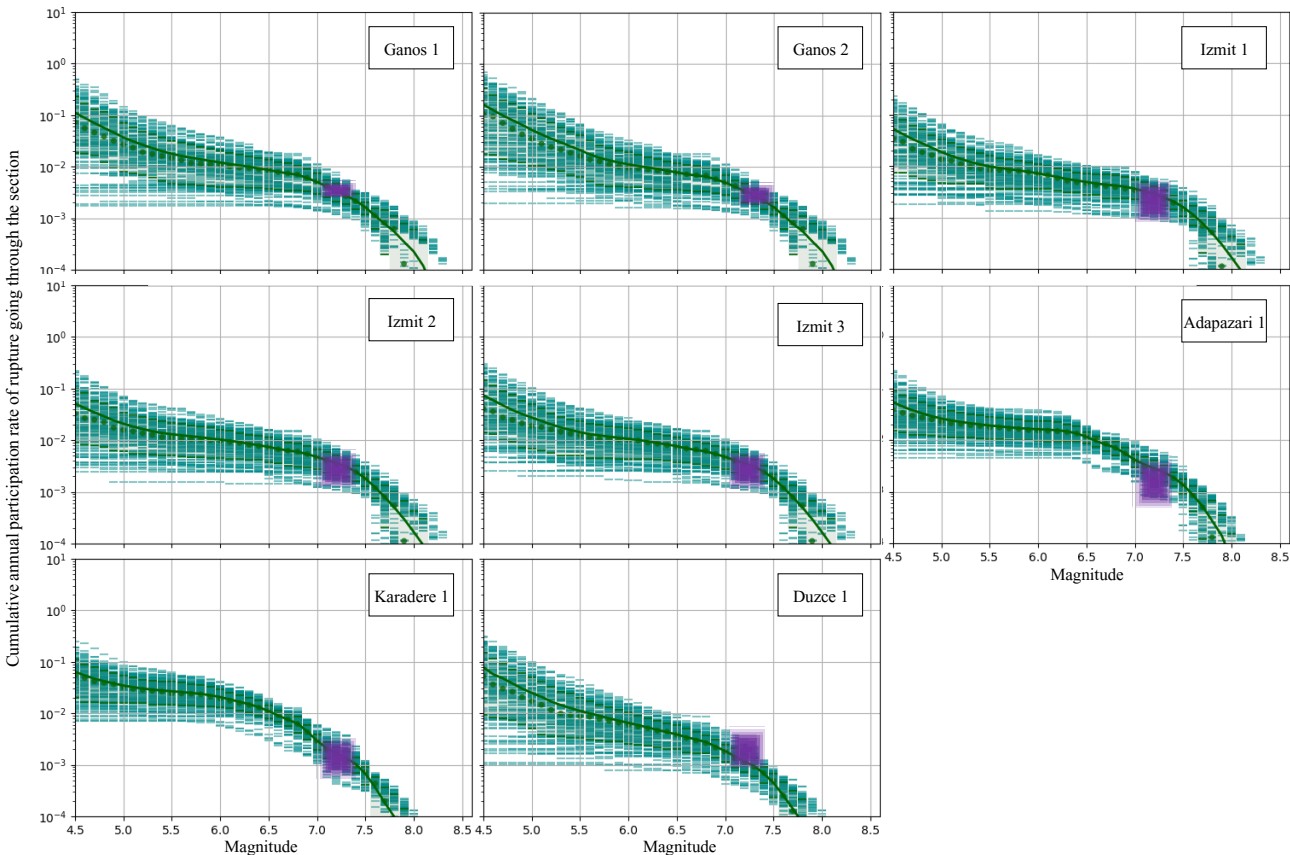

**Figure 8.** Comparison of the modelled rupture rates with the rates calculated from the paleoearthquake record at each paleoearthquake site (Figure 2). The green curves are the modelled rates of ruptures rupturing the given section. The dashed lines are each individual model of the logic tree. The darker dashed lines are the 16th and 84th percentiles of the distribution and the continuous line is the mean value of the distribution. Paleoearthquake rates in purple (see Table 3).

In the hope of reducing these uncertainties, we implemented our fault system in the physics-based simulator RSQSim
(Richards-Dinger and Dieterich (2012)) and analysed the resulting synthetic earthquake catalogues. RSQSim is a boundary element model that applies the rate and state equation (Dieterich (1978)) where each element of a fault system can be in three different states: loading, nucleating the earthquake rupture or sliding. Elements interactions are considered to be quasi-static therefore neglecting the dynamic influence of seismic waves that would be generated by sliding elements. Where the elements are sliding, they are doing so at a constant slip-rate. These simplifications of the earthquake physics and the rate and state law
allow RSQSim to be computationally efficient and to generate long earthquake histories for a large fault network.

RSQSim takes as input the same information as SHERIFS concerning the fault parameters but doesn't require a target MFD shape nor a set of possible FtF ruptures. The MFD of the fault system and the ruptures will be deduced from the synthetic

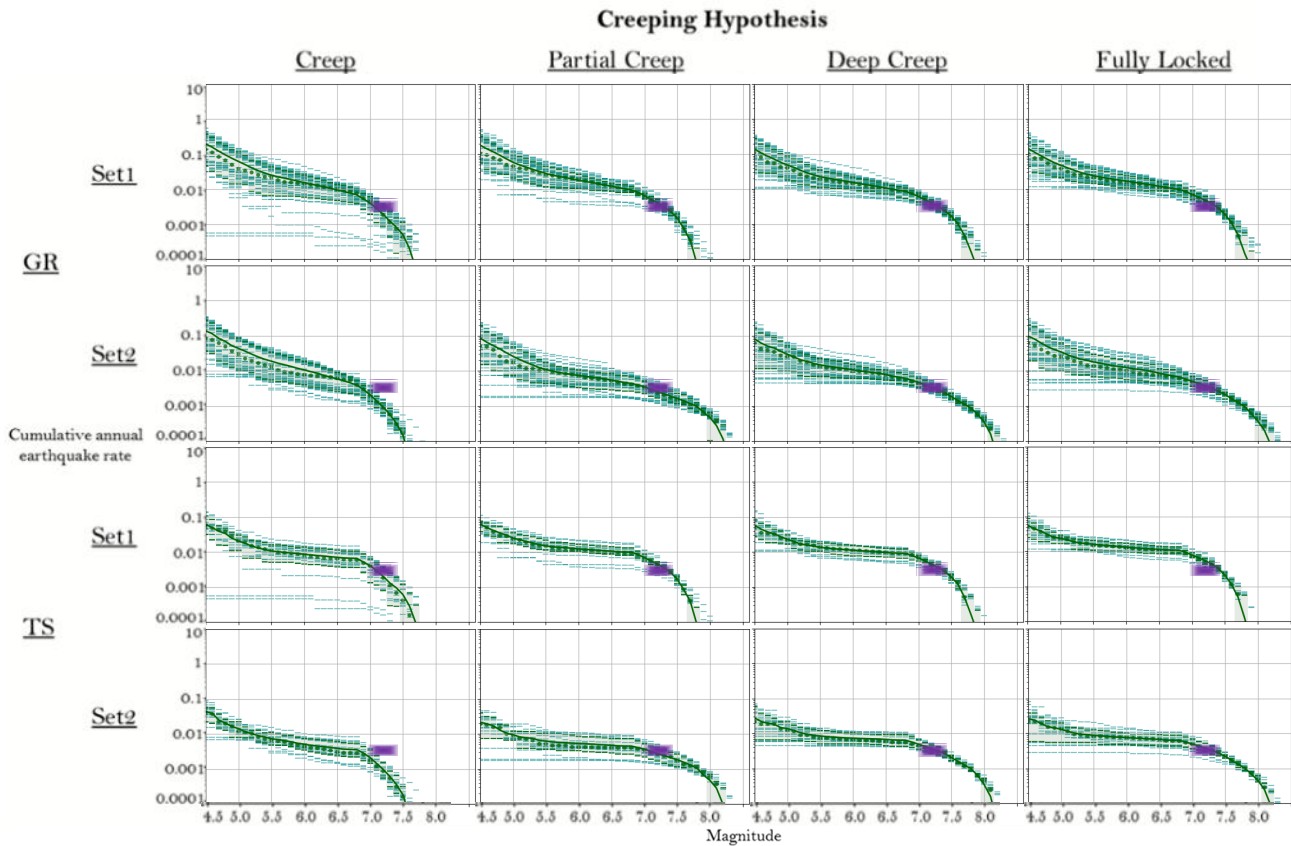

**Figure 9.** Comparison of the modelled rupture rates with the rates calculated from the paleoearthquake record at Ganos 1 site (Figure 2) for different branches of the logic tree. The green curves are the modelled rate of ruptures rupturing the Ganos segment. The dashed lines are each individual model of the logic tree. The dotted line are the 16th and 84th percentiles of the distribution and the continuous line is the mean value of the distribution. Paleoearthquake rates in purple (Table 3.

earthquake catalogue which results from the combination of the faults loading rates and the rate and state friction law as implemented in RSQSim.

In this study, we will use the same friction parameters a and b for all the elements of the fault system (0.01 and 0.015 respectively). These a and b value are based on empirical measurements from laboratory earthquake experiments and have been widely used for modelling earthquake cycles with the rate and state equation (Marone (1998)). The stress loading rate of the faults in the system is defined using back-slip loading. Back-slip loading calculates the stress rate for each element of the fault that corresponds to the long-term slip-rate of the fault.

We are using triangular elements of 1 km size. For this reason, we will only discuss earthquakes larger than magnitude 6 that rupture a number of elements large enough (around 100) to be representative of an earthquake rupture.





### 3.2.2 Earthquake rates in RSQSim

Using the fault geometry and slip-rate with the fully locked hypothesis (table 1, figure 2) as input for RSQSim, we simulate a 10 000 years long earthquake catalogue from which we remove the first 1000 years of seismicity to let the fault system run
through several earthquake cycles before starting the statistical analysis.

Richards-Dinger and Dieterich (2012) compared RSQSim with fully dynamic earthquake simulations and have shown that both modelling approaches lead to similar earthquake ruptures in terms of slip history and final slip on the fault. We observe that the scaling of earthquakes in RSQSim is in good agreement with the scaling laws (Tullis et al. (2012)) in terms of geometrical scaling and of stress drop. Interestingly, RSQSim reproduces quite well the rupture extent and magnitude of the 1999 of Izmit
and Duzce earthquakes. Based on these comparisons we conclude that even if the physics of earthquake is simplified to allow for cost efficient calculations, the simulated catalogue is representative of the natural system at least to the first order. In order to not over-interpret the results we limit our use of the RSQSim simulated catalogue to the discussion of two first order questions : (1) does the long term MFD of the fault system follow a GR law?, (2) are magnitude 8.0 earthquakes possible in this fault system?
Considering the MFD of the synthetic catalogue, we can reach two conclusions: RSQSim cannot reproduce a GR MFD with the given fault system and the maximum magnitude that can be generated is closer to magnitude 7.7 than to magnitude 8.0. These two conclusions are not affected by uncertainties on a and b values (figure 10).

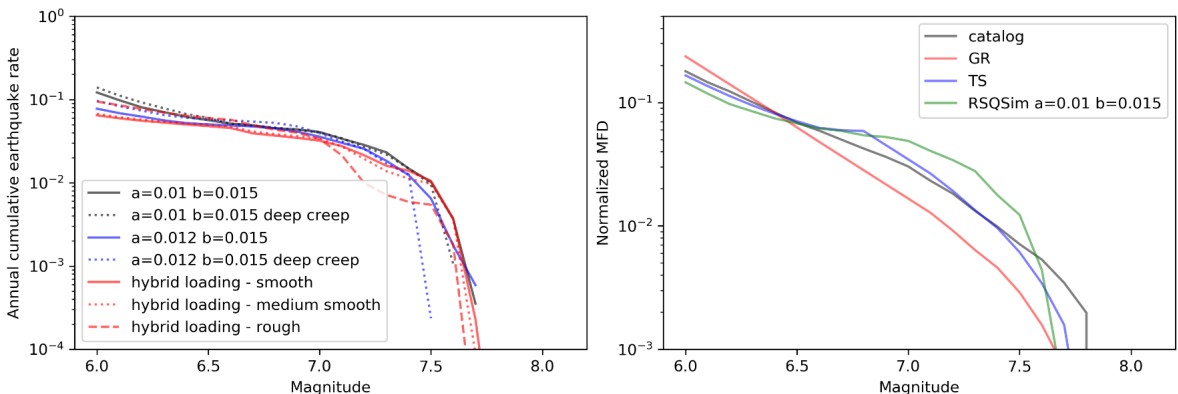

**Figure 10.** Left : MFD of the synthetic catalogues generated from the RSQSim simulations exploring a range of input parameters. Right : Normalised MFD of the earthquake catalogue, the MFD of the faults modelled using SHERIFS with a GR MFD or a TS MFD (the background seismicity has been removed), and the RSQSim synthetic catalogue (a=0.01 b=0.015).

We also explored the impact of having a region of the fault creeping in the Western Marmara Sea. In a way similar to the "deep creep" model used in SHERIFS, in RSQSim, elements that are below 5km depth were attributed a b value much lower
than the a value. The resulting MFD (figure 10) leads to the same first order conclusions as for the fully locked hypothesis. This is to be expected. Indeed, the region where creep is suspected is of a negligible size compared to the entire fault system





and does not drastically affect the shape at the system level MFD. The creeping condition of the fault in the western Marmara sea most likely affects the participation rates of neighbouring faults, however, here we only consider the first order results of the calculation.

The results are also compared with the hybrid loading method used in Shaw et al. (2018) with the same input parameters as in the Californian application. The hybrid loading method allows to remove the edge effects on the stress field of the fault that occur with the standard back-slip loading (Shaw et al. (2018)). Different smoothness parameters are tested. The synthetic catalogues do not follow GR MFD, strengthening our previous observation.

## 4    Discussion on the weight of the epistemic uncertainties in the logic tree

The flexibility of SHERIFS for modelling the earthquake rates in the NAFS allowed exploring a wide range of uncertainties in a logic tree framework. The SHERIFS approach doesn't contain any physical constraints; it can therefore accommodate the different input hypotheses that are being discussed by the scientific community but also can lead to a large range of uncertainties on the earthquake rates. Since SHERIFS only uses the fault data as input, it is possible to compare the modelled rates with rates calculated from the earthquake catalogue and paleoearthquakes which can be considered as independent data from the SHERIFS input.

Branches weights in a logic tree are usually based on the scientific value of each hypothesis explored in the logic tree but not on the capacity of the modelled earthquake rates in each branch to reproduce the data. The weight of an individual branch of the logic tree is simply the product of the weights of each hypothesis used for the branch. In this discussion, we propose an innovative approach to set the weight of each branch of the logic tree accounting for both the input hypotheses used and the ability to reproduce the independent data.

We set up a quantitative scoring system in order to set the weight of each individual branch of the logic tree. For each branch, four scores are calculated (figure 11). One score judges the ability of the combination of input parameters and hypotheses to spend the slip-rate budget of the faults as earthquake rate and not generate a large amount of NMS (S1). Two scores compare the model to the data, judging the capacity of the modelled earthquake rates to reproduce the earthquake rates deduced from the catalogue (S2), and to reproduce the earthquake rates deduced from the paleoseismic records (S3). The last score rates a-priory the input hypotheses based on the analysis of the RSQSim synthetic catalogue (S4). The final weight of each branch of the logic tree is given by the combination of the four scores (figure 11). The code allowing the computation of the scores and the excel file showing the computation of the weights are provided in the electronic supplements.

### 4.1    Uncertainty on the creep

#### 4.1.1    Scoring based on the ration of NMS

We calculate the NMS value for each model and each fault section of the central zone. Since a large NMS value is likely linked to incompatibilities between input hypotheses, given the SHERIFS framework, a low score is attributed to models with high

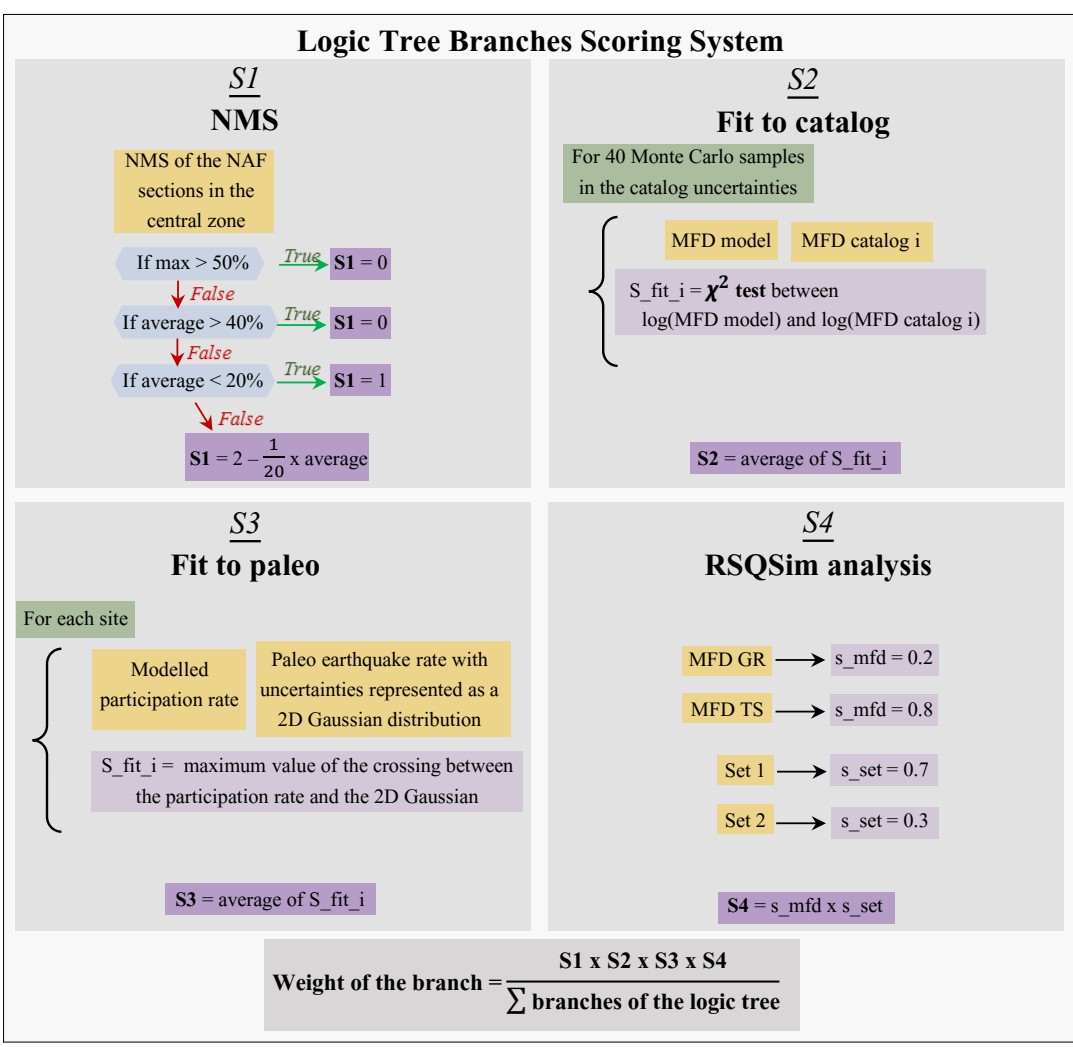

**Figure 11.** Scoring system allowing the individual weighting of each branch of the logic tree according to both its capacity to spend the fault slip rate budget into earthquake rates (S1), its capacity to reproduce the earthquake rates observed in the data and the agreement between the input hypotheses (S2 and S3) and the results of the discussion on the physics-based synthetic catalogue (S4).



NMS slip value. For a given model, if the mean NMS value for the fault sections of the central zone is greater than 40% or if the NMS value of one of the faults of the central region is greater than 50%, the score is zero. The score is 1 if the average NMS of the sections of the central region is less than 20%. Between the average values of 20% and 40%, the score linearly decreases from 1 to 0 as the NMS value increases. As a result, the average score of the fully creeping models is 0.27, the average score of the partly creeping models is 0.6 and the average scores of both the deep creep and fully locked models are 0.96.

The modelling of the creep conditions assumed on the Terkidag section play a predominant role in the modelling of earthquake rates on the adjacent Ganos section and hence the scoring based on NMS. The models considering the Terkidag fault as completely creeping have difficulties reproducing the paleoearthquake rates estimated on the Ganos fault (figure 9) and lead to a high level of NMS on this fault. In the fully creeping model, the Ganos section is only able to convert 54% of its slip-rate in seismic moment rate and 46% is considered NMS slip. In the partly creeping model, the Ganos section spends 71% of its slip-rate as moment rate. In the deep-creep model and the fully locked model, the Ganos section spends 90% of its slip-rate as seismic moment rate.

### 4.1.2 Scoring based on the fit to the rate of the paleo-earthquake

Figure 9 exposes the underestimation of the rate of earthquakes on the Ganos section compared to data when considering the Creep branch of the logic tree. The comparison with the paleoseismicity is done at all sites where paleoseismicity is available for each model of the logic tree (see above). At each site, the rate of paleo-earthquake is extracted from the literature (table 3) and represented as a 2D Gaussian distribution in order to capture the uncertainty both in terms of magnitude and annual rate of occurrence. At each site, the modelled rates are compared to this distribution. The closer the participation rate for a given section is from the maximum of the paleoearthquake rate distribution the larger the score. The score of the branch is the average score for each paleo-site. The average scores on the paleoearthquake fit for the Creep, Partial Creep, Deep Creep and Fully Locked models are 0.65, 0.73, 0.82 and 0.81 respectively.

### 4.1.3 Scoring based on the fit to the earthquake catalogue

The average score for models imposing a TS MFD are only slightly higher than those with a GR MFD as a target shape for the MFD of the fault system (0.52 versus 0.49). While both shapes show a good agreement with the rate of large earthquakes calculated from both the catalogue and the paleoseismicity, the models with a GR MFD overestimate the rate of small to intermediate earthquakes, as seen in Figure 7. Given the uncertainty on the annual rates calculated from the earthquake catalogue, the general better fit obtained when using the TS MFD is only leading to a slightly stronger score for this branch compared to the GR MFD.

### 4.2 Scoring based on RSQSim results - MFD

It can be argued that the deviation of the MFD shape from the GR shape is an artefact of the observation period, which is too short for accurately capturing the full earthquake cycle. Such arguments have been brought forward in California in support





for a GR MFD target shape even when the apparent shape of the earthquake catalogue close to the faults might differ from the
GR shape (Page and Felzer (2015)). Stirling and Gerstenberger (2018) have shown that for four fault systems in New Zealand,
while the data doesn't show a good agreement with the GR MFD hypothesis, ETAS simulations suggest that the observation
time in the data is too short to accurately calculate the long term rate of a clustered earthquake catalogue.

However, the 10 000 years long synthetic catalogue generated by RSQSim, complete and representative of the seismic cycle
by definition, also diverges from the GR MFD shape. In the RSQSim catalogue, there are almost no earthquakes of magnitude
between 6.5 and 7.0. According to the scaling laws (Thingbaijam et al. (2017)), 6.5 is the magnitude of an earthquake rupturing
the whole fault width. We speculate that in RSQSim, an earthquake rupturing the entire width of the fault, is more likely to
rupture neighbouring faults unless stopped by geometrical complexities or the presence of faults that are still in the early part
of their cycle. The average length of the straight sections of the faults is around 100 km (figure 1), which corresponds to an
earthquake of magnitude around 7.2. Ruptures are able to overcome bends and asperities if the neighbouring fault is close to
rupture leading to larger magnitude earthquakes. Since they require several sections to be close to rupture simultaneously, these
ruptures are less frequent. We obtain therefore a bi-modal distribution with the earthquakes up to magnitude 6.3 following a
GR, rupturing different portion of the fault plane, very few earthquakes between magnitude 6.3 and 7.0, and a population of
earthquakes larger than magnitude 7.0 with the number of earthquake decreasing with the magnitude.

Based on the comparison between the modelled rates with the rates of the earthquake catalogue and the results of RSQSim,
and in consideration of the scientific debate around the MFD, we suggest a stronger score (s_mfd) on the "Tuned shape" MFD
branch. The branch exploring the "GR MFD" should remain part of the logic tree, however. Consequently, we set the score of
the branch "Tuned shape MFD" to 0.8 and the score of the branch "GR MFD" to 0.2.

### 4.3 Scoring based on RSQSIM results - Mmax

Based on the comparison between the modelled rates from SHERIFS and the rate calculated from the earthquake catalogue and
the paleoseismicity studies, it is not possible to weight differently the two branches of the logic tree exploring the uncertainty
on the rupture scenarios. While the fit between the modelled rates and the catalogue rates using the Set 2 of ruptures, allowing
larger ruptures, leads to a slightly better fit than the models using the Set 1 of ruptures, both fits can be considered as satisfying
(figure 7). Whether ruptures up to magnitude 8.0 are considered or not, the rates in the models are matching the rates from the
data when using the TS MFD as a target.

In the several 10 000 years long catalogues simulated by RSQSim, we do not observe earthquake larger than 7.7. In the
SHERIFS models, earthquake with magnitude larger than or equal to 8.0 have an annual frequency of 5 10-4. The likelihood of
observing at least one earthquake in a 10 kyr long catalogue is 99.3 %. While the bends and other geometrical complexities of
the fault trace are crossed by ruptures in the model, no rupture crosses enough complexities to generate such large earthquakes.
We can assume that the fault system is very chaotic and having a large number of neighbouring fault sections close to rupture
is very unlikely. However, it is important to recall that RSQSim does not fully model the dynamic effects of the earthquake
rupture. It is possible that this simplification in the modelling leads to underestimating the likelihood of a rupture propagating





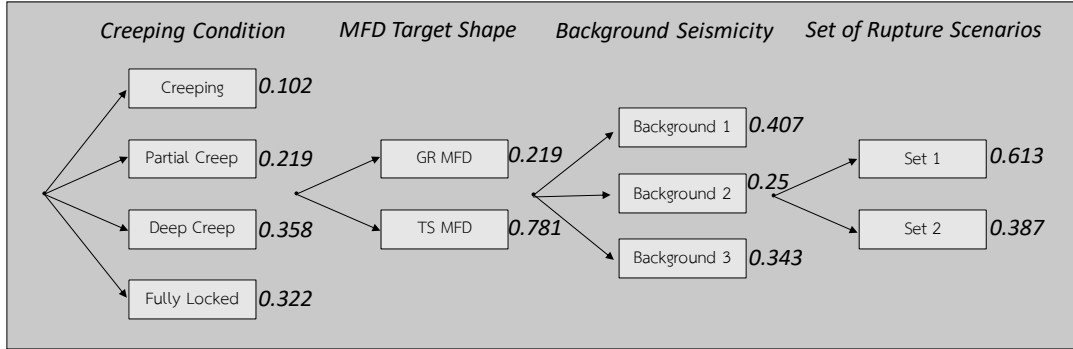

**Figure 12.** Weighted logic tree established in this study. For each branch, the scaling law parameters and the slip-rate uncertainties are explored through 10 random samples. The weight of each branch is indicated by the bold number.

through fault complexities. In consideration of these arguments, we set the s_set score of the branch "Set 1" to 0.7 and the s_set score of the branch "Set 2" to 0.3.

## 4.4 The weighted logic tree

In the discussion, we established four types of scores for the logic tree branches: the weights established from the analysis of the RSQSim synthetic catalogues and those established from the comparison of the earthquake rates modelled using SHERIFS with the rates calculated from the earthquake catalogue and the paleoearthquake record. For each individual branch of the logic tree, these scores are convolved into a final weight unique to the branch (figure 11).

   The final weight of each hypothesis is calculated by summing all the branches using a given hypothesis (figure 12). Due to
370 the high NMS value of the models using the Creeping hypothesis and their underestimation of the paleoearthquake rates, this hypothesis has the smallest weight in the logic tree.

   The weights of the MFD hypothesis branches and the Set of Rupture scenario branches are strongly influenced by the scores imposed after the analysis of the RSQSim synthetic catalogue. While the fit to the catalogue is better with the TS MFD hypothesis, the fit to the paleoearthquake rates and the ratio of NMS is similar for both branches. Therefore the comparison
with the data affects the weight only marginally. The same can be said about the set of rupture scenario branches.

   The weights of the background hypothesis branches are only affected by the scores depending of the comparison with the data. Overall, the background 1 hypothesis reproduces the earthquake rates better than the other two background hypotheses. In figure 7, we can observe that the modelled MFD tends to be slightly higher than the catalogue MFD for the majority of models. Therefore, the background 3, adding more seismicity to the background than the two other hypotheses (table 4) necessarily
leads to earthquake rates that are in general slightly higher than the observed rates. It is worth noting that we are discussing

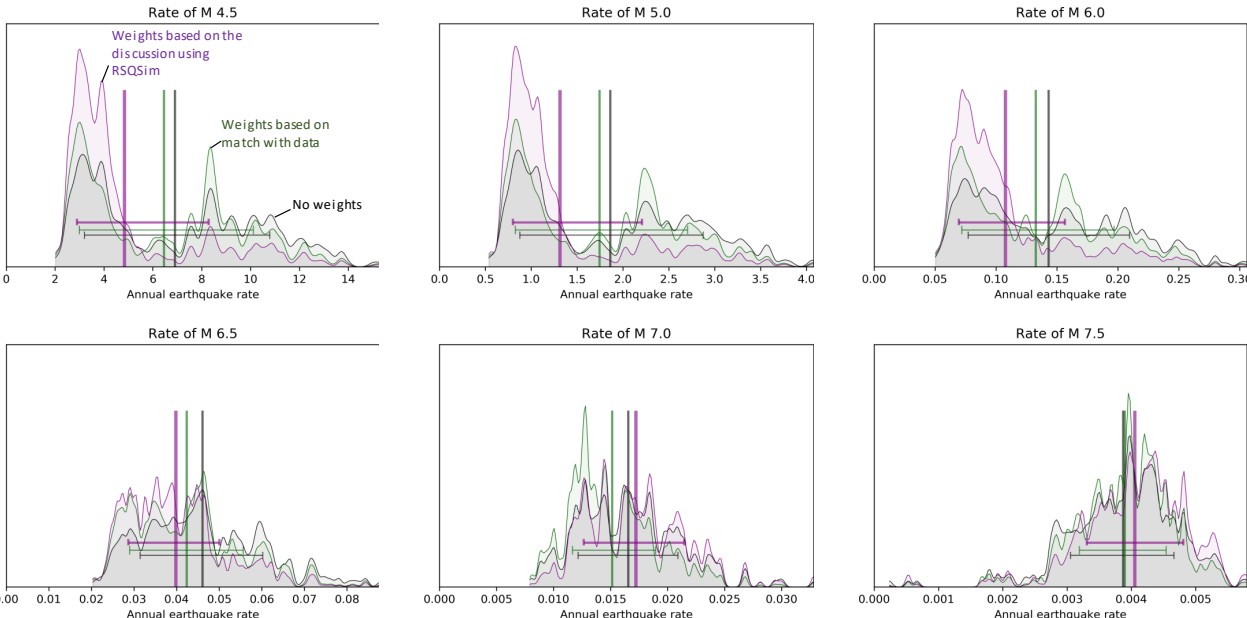

**Figure 13.** Distribution of the cumulative earthquake rates in the logic tree for the central zone for different magnitudes. In black : the density distribution when only the scoring according to the NMS is applied. In green : the density distribution weighted according to the fit with the data (catalogue and paleoseismicity). In purple : the final distribution of earthquake rates, weighted according to the match with the data and the discussion based on RSQSim. The vertical bar shows the mean annual earthquake rate of each distribution. The horizontal bar extends between the 16th percentile and the 84th percentile of each distribution.

small differences in trends that are correctly captured by the weights in the logic tree with the three hypotheses having similar weights with a slightly stronger weight for background 1.

The impact of the weights on the annual earthquake rates are presented in figure 13. The weighting leads to lower earthquake rates for the small and intermediate magnitudes and larger rates for the magnitudes greater than 7.0. This is due to the reduced weight of the branches using the GR hypothesis.

## 5 Conclusions

In this study, we calculated earthquake rates in the Marmara region relying on the fault slip-rate and geometry as primary information. We combined two innovating approaches, the SHERIFS approach that relies on statistical rules and the RSQSim approach that relies on physical rules.

With SHERIFS, we explored an extensive logic tree of uncertainties concerning the locking condition of the NAF fault in the Marmara region, the shape of the MFD, the ratio of seismicity between the background and the faults, the largest possible rupture as well as uncertainties on the slip-rates and the maximum magnitude predicted by the scaling law. Rather





than basing the weights of the branches of the logic only on expert judgement of each hypothesis, we rather take advantage of model performance by comparing results with data (earthquake catalogue and paleoearthquake) and weighting the branch

accordingly. In addition, the analysis of the synthetic catalogue simulated by RSQSim showed a MFD diverging from a GR and no earthquake of magnitude larger than 7.7. This allowed us to attribute a stronger weight to the branches of the logic tree showing similar features leading to a final weighted distribution of modelled annual rates that properly represents the state of knowledge of the NAFS in the vicinity of Istanbul (figure 13).

In a companion article, we carry out a Probabilistic Seismic Hazard Assessment using the weighted logic tree and calculate

the risk of collapse of a building in Istanbul. We discuss the impact of each uncertainty of the logic tree on the probability of collapse and we deaggregate the risk results in order to identify the sources controlling the seismic risk.

*Code and data availability.* All the data and input parameters used for the earthquake rate calculation with SHERIFS are available in the electronic supplements.The SHERIFS code is available at https://github.com/tomchartier/SHERIFS and in the electronic supplements.

*Acknowledgements.* This project was partly founded by the Axa Research Fund. We thank Aurelien Boiselet from Axa for his involvement in the project.



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
