# Peer review of "Modelling earthquake rates and associated uncertainties in the Marmara Region, Turkey"

_Natural Hazards and Earth System Sciences, 2020_

## Author Comment (AC1)

General response

We thank both reviewers for their valuable comments, remarks and questions. We have answered below to each remark and brought the appropriate changes to the manuscript.

**Reviewer 1**

The work by Chartier and collaborators present a statistical approximation to model earthquake rates in the Marmara region in order to be used as input for a PSHA based on fault rates (a companion paper). They use the SHERIFS code to model the seismicity rates on the North Anatolian Fault (NAF) system in the Marmara region and apply a logic tree approach to weight different parameters of the model. The weights are based on the model performance, and they use a physics-based earthquake simulator (RSQSim) to obtain weights for some of the tree branches. The results are probability density functions for earthquake rates of different magnitudes.

The work is interesting as the hybrid statistical-physical approach is not common and the use of earthquake simulators or earthquakes rates linked to the geological characteristics of faults, is an approximation to the PSHA rapidly growing.

The work by Chartier and collaborators deserves to be published and fits perfectly on the scope of NHESS. However some revisions are needed as is detailed below.

Main concerns:

The use of RSQSim by the authors in this work is rather basic. This code, as any physics-based code, needs some adaptation and tweaking to the modelled system in order to represent confidently an approximation to the natural behavior. The developers of RSQSim are between the authors of the paper, and they are aware of this, which makes me wonder how deep has been their involvement in the developing of this work. Variations in a, b and Dc values of the rate-and-state friction model implies variations in the earthquake nucleation process, events clustering and major earthquakes frequencies. The characteristics of the 3D geometrical model, the presence of complexities and fault overlaps, are key in the earthquake rupture propagation. Other physical parameters (stresses, mechanical constants) has also influence on the final earthquake catalog produced.

The authors use RSQSim as control model in order to weight the branches of maximum magnitude and shape of the FMD. In my opinion, in view of the simplistic approximation used for RSQSim, their results are not robust enough to be used as control, and on the contrary increases the uncertainty on the results.

I think the authors should made a decision, they can drop the RSQSim part, weighting the maximum magnitude for example with the historical observations, and relying the work on the SHERIFS code; or they can increase the effort on the RSQSim part, exploring different values for the key parameters and adjusting the model behavior to fit it to the instrumental catalog; exploring the same hypotheses explored with SHERIFS. If the

authors choose to do the latter then they can compare the results from both approximations discussing the limitations and advantages of both methodologies.

In the RSQSim model they use standard frictional parameters from Marone (1998), but estimations of the parameters of the rate-and-state model for the area could have been used instead (e.g. Kaneko et al., 2013). Also they state that only events greater than Mw 6 are representative, but in previous works models with similar characteristics have shown to adequately represent FMD for events with magnitudes over 5 (e.g. Dieterich and Richards-Dinger, 2010; Shaw et al., 2018). This decision seems rather arbitrary and should be backed by some analysis.

In general the RSQSim part seem scarce, for example in line 253 the authors state that "the scaling of earthquakes in RSQSim is in good agreement with the scaling laws (Tullis et al. (2012)) in terms of geometrical scaling and of stress drop" but they do not show any proof of that, specially taking into account that this fit depends on the parameters used in the model.

At the end the paper seem cut, although the results are presented and somewhat discussed along the text, there is no discussion section and the conclusions are brief. As mentioned before I suggest to expand the RSQSim part and discuss the limitations, advantages and differences between the two models and the results obtained. The other option is to drop the RSQSim part and discuss the differences on the obtained results with or without the weighted logic-tree. A hazard map with the results would be great but I suppose that it is included in the companion paper.

Answer to the main concerns:

The reviewer general concern highlighted a lack of clarity in the manuscript on the use of RSQSim our earthquake rate modelling process.

The goal of the study is not to compare the results of the two modelling approaches (SHERIFS and RSQSim) in a parallel manner but to bring additional information to the modelling of the earthquake rate using the physics-based simulator, aiming to reduce the uncertainty affecting the earthquake rate estimate and down the line, the hazard assessment. We did note try to fit the results of RSQSim and SHERIFS. We kept the two approaches independent as much as possible, only using the same fault information as input. For the input parameters of RSQSim, we use general values and did not try to modify them to fit the results of SHERIFS as it would counter the aim of the study and not bring additional information. We agree with the reviewer that more exploration of the parameters of RSQSim could be possible but the goal of our study is to bring the physics-based models closer to the hazard models. Developing a fully explored physics-based model is out of the scope of any hazard study, however, we think that a sufficiently explored RSQSim can already bring valuable information that can help to weight different hypotheses in the logic tree, in parallel with the comparison with local data.

To answer the reviewer's comment on the discussion part, we have clarified the titles in section 4 of the paper to emphasize that the weighting of the branches of the logic tree is itself a discussion of the different hypotheses explored in this study. We also have

improved the discussion and conclusion parts of the manuscript clarifying our use of RSQSim in this study.

Concerning the agreement between the modeled earthquakes and the scaling laws, we agree with the reviewer's comment that this fit is dependent of the input parameters used for RSQSim. In figure 1R, we can see that for the parameters used in the study (a = 0.01 and b=0.015) the agreement between the model and the scaling law of Wells and Coppersmith 1994 is acceptable for magnitudes up to 7.3. For larger magnitudes, the rupture areas are smaller than predicted by the scaling law but remains within the standard deviation of 0.23 published in Wells and Coppersmith 1994 for this scaling law linking rupture area to magnitude.

[Figure]

*Figure 1R : Scaling of the rupture area with the magnitude for each simulated earthquake in RSQSim with parameters a= 0.01, b = 0.015 and normal stress of 100 MPa. In red, the scaling relationship from Well and Coppersmith 1994.*

For the sake of the length of the article, we have decided to keep the structure of the manuscript as it is and leave the hazard and risk calculation in the companion paper. Since this companion paper is not yet ready to be submitted, we changed the way we refer to it in the introduction (line 72) and the conclusion (line 429).

Other observations:

Figure 2. The triangles are purple and not yellow as stated in the caption. I suggest to move this triangles to the map a) in order to relate them easily the paleoseismic sites listed in table 3. Although the study area is well known to any earthquake-related researcher the maps should show the geographic coordinates.

The color in the caption has been corrected and the location of the paleoseismicity studies have been moved to fig 2a.

The reference to Woessner et al. (2015) in table 2 should be Stucchi et al. (2013) according to the reference in line 110.

We are referencing the SHARE project for the estimation of the completeness periods. (line 111)

Figure 3. The black squares seem gray to me and hard to see.

We would like to avoid to focus the attention on the mean value but rather on the range of the earthquake rate, especially for the larger magnitudes. The color in the caption has been changed to grey (figure 3).

Most of references in parentheses present wrong format, the year should be out of parentheses and after a coma.

The template used is the one provided by the journal. We will exchange with the editor to ensure the correct format is used.

I think that the earthquake rate from paleoseismicity is wrongly calculated in table 3. In Paleoseismology the observation time is usually the time between the first event (or the older geological unit) and the present. Specially when there are a few events the number of "inter-event times" observed is not the same of the number of events, as the first event is observed only by the coseismic rupture, and consequently the earthquake rate must be ER=(n-1)/OT; the same reasoning applies when the last earthquake occurred is very recent, as the case of Izmit 1999, event. Please revise the calculations taking this into account and correcting the numbers when necessary.

In this study, we use the paleoseismicity in a similar fashion as the earthquake catalog to discuss modelled earthquake rates. We decided to use the observation period of each trench and the number of events observed to calculate an average rate rather than focusing on the inter-event times. In this manner, the average event rate is more relatable to the Poissonian earthquake rate considered in the PSHA calculation. For the site with a low number of events, the uncertainty on the rate is larger according to the Poissonian approximation.

We have added some clarification in the text (line 119).

Line 151. "In most PSHA, this is taken into account by a background zone with a GR MFD truncated at a given Mt." Does the authors have any reference to this assertion?

A reference to SHARE was added as an example. (line 155)

In line 155 the authors state that "… we only consider the instrumental catalogue after 1970." Although it is not clear to me the way the instrumental catalog is used to estimate the proportion of background seismicity.

The spatial distribution was used to broadly inform the different hypotheses explored for the background. Unfortunately, there is no method for estimating properly these value at the moment. In this study, we explore three hypotheses, acknowledging the uncertainty on the share of seismicity between the background and the faults, this uncertainty being for the moment ignored totally in the wide majority of hazard studies which rely on a truncating magnitude.

We have added some clarification in the text (lines 157 and 161).

Line 160. The phrase is not ended.

The sentence is completed. (line 165)

Line 174. Rather than "To reflect the lack of consensus…" the authors could say "To consider the alternative hypotheses…"

We modified the text. (line 178)

I have doubts on the formulation shown in page 11.

The first equation shown corresponds to the Cosentino expression of a truncated Gutenberg-Richter relation where a maximum magnitude for the entire catalog is assumed. In this case the maximum magnitude used is 6.8, while the maximum magnitude proposed for the entire catalog is 7.7 or 8.0. Also the use of "m" is confusing, does the authors mean "M"? If not explain what "m" means. Pi(m=6.6)/3 means Pi(M)/3 for M=6.6 using the truncated Gutenberg-Richter expression? The Pi(m=6.3) is in fact Pi(m=6.8)?

We have clarified the equation. (line 177)

The sections 3.1.5 and 3.1.6 are too short. Two phrases each. Please consider rearrange this paragraphs into other sections.

We rearrange the two sections. Adding some information to the first and merging the second to the results section. (line 203)

The lines 100-104 could be part of the same paragraph of section 3.1.5.

We decided to leave the presentation of the locking hypotheses with the data part of the manuscript and reference the table in section 1.5.

In figure 5 why there are duplicated fault traces? Does that mean that rupture twice in the same event? Are they different events? If so, consider use different color per rupture and the addition of the corresponding magnitudes as label for each.

We have duplicated the fault traces to show that a given section can rupture in several rupture scenarios. We have added some clarification in the text (line 194) and in the caption of the figure.

Line 210. Errata "ruptures est fit".

Corrected to « best » (line 218)

The title of section 3.2.1 is not needed and could be simply the first paragraphs of section 3.2.

We rearranged this section. (line 234)

Line 283-284. In fact the weighting based on performance is not innovative as has been used and discussed before (e.g. Scherbaum and Kuehn, 2011; Delavaud et al., 2012).

These studies are discussing the weighting of GMPE logic trees. We have changed the manuscript to precise that we discuss the weight of the earthquake source models. We change "innovative" to "novel". (line 295)

Errata on section 4.1.1 title "ratio" instead of "ration"?

It was a typo. It has been fixed. (line 306)

Scoring based on paleo-earthquakes. Please revise the rates obtained by paleoseismology as indicated before.

Referring to our answer before, we decided to keep the calculated rates. (line 119)

The scoring system of the logic-tree branches is not clearly explained. On figure 11 appear 4 items to compute the weights, and are explained along the text. But the logic tree presented on figures 6 and 12 (both figures are redundant and could be presented just the 12 with the weights) present different items and the weights shown differ from the values mentioned in the text with the other 4 criteria established. It is elusive to me how the computation of the weights is finally done.

Since the weights are attributed after the calculation of the earthquake rates and comparison to the data, we think both figures are necessary in order to not have the weights shown before the discussion since in hazard studies, the weight are usually set a priori, we do not want a hazard modeler reader to assume that the weights are fixes a priori here while reading the manuscript.

The final weights values are the combination of the score as explained in figure 11. The full weight calculation is available only in the supplementary materials as it would be cumbersome to do a full description of the weights in the main manuscript.

We have added a reference to the equation in figure 11 (line 393)

Line 356. Wrong formatting of the annual frequency, 5 10-4 has no sense. I suppose that it is $5 \times 10^{-4}$) but correctly formatted.

We changed the formatting (line 372)

Figure 13. The Y-labels should be present. Are they the values of the density function in probability between 0 and 1?

We have precised that they are density functions. Since the Y values of a density function is of little use for the reader, we have decided to lighten the figure. The area of a density function is 1 by definition. (figure 13)

References:

Delavaud, E., Cotton, F., Akkar, S., Scherbaum, F., Danciu, L., Beauval, C., ... & Theodoulidis, N. (2012). Toward a ground-motion logic tree for probabilistic seismic hazard assessment in Europe. Journal of Seismology, 16(3), 451-473.

Dieterich, J. H., & Richards-Dinger, K. B. (2010). Earthquake recurrence in simulated fault systems.Seismogenesis and Earthquake Forecasting: The Frank Evison Volume II (pp. 233-250). Springer, Basel.

Kaneko, Y., Fialko, Y., Sandwell, D. T., Tong, X., & Furuya, M. (2013). Interseismic deformation and creep along the central section of the North Anatolian Fault (Turkey): InSAR observations and implications for rateâ andâ state friction properties. Journal of Geophysical Research: Solid Earth, 118(1), 316-331.

Scherbaum, F., & Kuehn, N. M. (2011). Logic tree branch weights and probabilities: Summing up to one is not enough. Earthquake Spectra, 27(4), 1237-1251.

Shaw, B. E., Milner, K. R., Field, E. H., Richards-Dinger, K., Gilchrist, J. J., Dieterich, J. H., & Jordan, T. H. (2018). A physics-based earthquake simulator replicates seismic hazard statistics across California. Science advances, 4(8), eaau0688.

---

## Author Comment (AC2)

General response

We thank both reviewers for their valuable comments, remarks and questions. We have answered below to each remark and brought the appropriate changes to the manuscript.

**Reviewer 2**

General comment:

In this work, Thomas Chartier ET al. computed the earthquake rates in the Marmara region with two approaches: The SHERIFS and the RSQSim. By the first one, the authors model the earthquake rates and explore a logic tree of epistemic uncertainties regarding the locking condition of the fault. They combine this statistical approach to a a physical one by means of the simulator RSQSim, which inform the logic tree to obtain weights for the tree branches.

I appreciate the dual approach that helps surpassing the limitations of both methods when individually considered. since NHESS is focused on modeling natural hazards and this work matches very good the disciplines of the journal, it deserves to be published after minor revisions.

Specific comment:

Given the excellent performance of the method, I wonder if it MAY BE possible TO IMPROVE THE RESULT, perhaps by varying the parameters in the RSQSim simulator (rate and state parameters). In general, I think that the physics-based part gives an important boost to the final result as well as being one of the innovative parts of the work. A better discussion of the results and the highlighting of the actual improvement due to the physical approach could support the article as a whole.

We thank the reviewer for the comment.
We have improved our discussion and conclusion sections in order to better explain the use of RSQSim.

Technical corrections:
Line 72 -> I guess there is a missing reference.
We have modified the references to the companion study since it is not submitted yet (lines 72 and 429).
Caption Fig.2 -> I don't see the yellow triangles in figure.
Caption has been corrected. (fig 2)
Table 3 -> I suggest to adjust the number of significant digits if the uncertainty values, according to the ones used for the annual earthquake rate.
The number of significant digits has been corrected. (table 3)

Line 152 -> Mt is not defined before.

We corrected the formulation of the equation. (line 178)

Line 160 -> The sentence "The proportion of earthquakes considered to occur on the faults for each branch is presented in" must be completed.

The reference to the figure was added.(line 165)

Line 356 -> I don't understand the number 5 10-4. Please, explain further.

The format was corrected the $5.10^{-4}$ (line 372)